# Hepatitis C (HCV) therapy for HCV mono-infected and HIV-HCV co-infected individuals living in Nepal

**Sudhamshu KC**[1☯], **Holly Murphy**[2☯], **Sameer Dixit**[3], **Apurva Rai**[4], **Bickram Pradhan**[5], **Marie Lagrange-Xelot**[6], **Niyanta Karki**[1], **Amélie Dureault**[7], **Ujjwal Karmacharya**[4], **Santosh Panthi**[3], **Nabin Tulachan**[8], **Prawchan KC**[4], **Anjay KC**[4], **Rajesh Rajbhandari**[3], **Andrew B. Trotter**[9], **Jörg Gölz**[10], **Pierre Pradat**[11]*, **Christian Trépo**[12], **Philippe Creac'H**[13]

1 National Academy of Medical Sciences, Kathmandu, Nepal, 2 Saint Joseph Mercy Ann Arbor Hospital, Ann Arbor, Michigan, United States of America, 3 Centre for Molecular Dynamics Nepal (CMDN), Kathmandu, Nepal, 4 Society of Positive Atmosphere and Related Support for HIV and AIDS, (SPARSHA-Nepal), Kathmandu, Nepal, 5 B.P. Koirala Institute of Health Sciences, Dharan, Nepal, 6 CHU La Réunion, Saint Denis de la Réunion, France, 7 Centre Hospitalier de Valence, Valence, France, 8 Manish Care Foundation, Pokhara, Nepal, 9 University of Illinois at Chicago, College of Medicine, Chicago, Illinois, United States of America, 10 Praxiszentrum Kaiserdamm, Berlin, Germany, 11 Centre for Clinical Research, Croix-Rousse Hospital, Hospices Civils de Lyon, Lyon, France, 12 INSERM U1052, Lyon, France, 13 The Global Fund, Geneva, Switzerland

☯ These authors contributed equally to this work.
* pierre.pradat@univ-lyon1.fr

**Data Availability Statement:** All relevant data are within the manuscript and its Supporting Information files.

## Abstract

### Background

Despite direct-acting antivirals (DAA), aims to "eradicate" viral hepatitis by 2030 remain unlikely. In Nepal, an expert consortium was established to treat HCV through Nepal earthquakes aftermath offering a model for HCV treatment expansion in a resource-poor setting.

### Methodology/Principal findings

In 2015, we established a network of hepatologists, laboratory experts, and community-based leaders at 6 Opioid Substitution Treatment (OST) sites from 4 cities in Nepal screening 838 patients for a treatment cohort of 600 individuals with HCV infection and past or current drug use. During phase 1, patients were treated with interferon-based regimens (n = 46). During phase 2, 135 patients with optimal predictors (HIV controlled, without cirrhosis, low baseline HCV viral load) were treated with DAA-based regimens. During phase 3, IFN-free DAA treatment was expanded, regardless of HCV disease severity, HIV viremia or drug use. Sustained virologic response (SVR) was assessed at 12 weeks.

Median age was 37 years and 95.5% were males. HCV genotype was 3 (53.2%) or 1a (40.7%) and 32% had cirrhosis; 42.5% were HIV-HCV coinfected. The intention-to-treat (ITT) SVR rates in phase 2 and 3 were 97% and 81%, respectively. The overall per-protocol and ITT SVR rates were 97% and 85%, respectively. By multivariable analysis, treatment at the Kathmandu site was protective and substance use, treatment during phase 3 were associated with failure to achieve SVR.

**Funding:** Save the Children received a grant from the Global Fund (grant number NPL-H-SCF). The funder had no role in study design, data collection and analysis, decision to publish, or preparation of the manuscript.

**Competing interests:** The authors have declared that no competing interests exist.

## Conclusions/Significance

Very high SVR rates may be achieved in a difficult-to-treat, low-income population whatever the patient's profile and disease severity. The excellent treatment outcomes observed in this real-life community study should prompt further HCV treatment initiatives in Nepal.

### Author summary

Despite very effective antiviral therapies, Hepatitis C virus (HCV) eradication remains a major challenge, especially in resource-limited countries. In Nepal, which ranks among the poorest countries in the world an expert consortium was established to treat HCV patients in six centers throughout the country.

A cohort of 600 individuals with HCV infection and past or current drug use were treated using different treatment strategies over time. Very high treatment response rates were achieved in a difficult-to-treat, low-income population whatever the patient's profile and disease severity and despite the severe 2015 earthquakes in Nepal. The excellent treatment outcomes observed in this real-life community study should prompt further HCV treatment initiatives in Nepal.

## Introduction

Chronic hepatitis C virus (HCV) infection is a major public health problem worldwide, affecting more than 71 million people and responsible for 400,000 deaths per year [1]. Treatment of HCV infection has undergone a recent revolution since the advent of direct-acting antiviral (DAA) agents achieving virological cure in more than 90% of patients [2] with simplified treatment regimens with increased tolerability and efficacy compared to prior regimens of interferon (IFN) and ribavirin. These new therapies brought optimism about potential eradication of HCV, and in 2016 the World Health Organization (WHO) launched a program to "eradicate" viral hepatitis by 2030 [3]. In low-income countries, the price of HCV treatment and lack of diagnosis remain obstacles to the broad implementation of DAA regimens.

Nepal is a small South Asian country located between India and China with a population of 29 million inhabitants. It ranks among the poorest countries in the world. In the mid-1990s, the HCV seroprevalence in the general population was estimated at 0.6% [4] and in 2008, 0.66% [5]. Though prevalence estimates in the general population are low, they are higher among IVDU with a prevalence of detectable HCV RNA of 42% [6]. In 2016, an estimated 130,000 individuals were infected by HCV in Nepal [7]. Sadly due to a lack of awareness and health infrastructure and poverty, only a small fraction of the infected population has access to treatment [7].

Until 2014, there was no national program for surveillance, treatment, and prevention of HCV in Nepal. In May 2014, an Expert Consortium proposed to validate short-course IFN-based HCV treatment in Nepal based on data suggesting favorable characteristics of the population for response to IFN-based therapy [6]. We conducted a prior study demonstrating predominantly HCV genotype 3 (59.8%) along with HCV genotype 1 (40.2%) [6]. At that time, IFN and ribavirin were the only approved drugs in Nepal for HCV treatment and the DAAs remained out of reach. We developed guidance for immediate HCV treatment using shortened treatment with pegylated (PEG)-IFN and ribavirin when baseline HCV RNA level was low

and when rapid virological response (RVR) could be achieved. With the support of Global Fund and Deutsche Gesellschaft für Internationale Zusammenarbeit (GIZ) GmbH (German Development Cooperation) and through Save the Children, we provided free access to HCV treatment for 600 patients. Once the DAAs became available in Nepal, we revised the protocol and obtained DAA regimens. In this paper, we describe this large-scale treatment initiative with regard to patients and HCV characteristics and treatment outcomes for those treated with DAAs. Those treated with IFN-based regimen during phase 1 are described in S1 Appendix and S1 Table. During the protocol period, Nepal faced the severe Gorkha earthquake on April 25, 2015 with subsequent quakes and aftermath. There were nearly 22,000 people injured and 9,000 people killed. All study sites were within the areas affected by the earthquakes.

## Material and methods

### Ethics statement

This study was approved by the Ethical Committee of Nepal Health Research Council (NHRC). All patients gave their written inform consent.

We created a collaborative research network between existing hepatologists, laboratory experts, and existing community-based opioid substitution therapy (OST) and HIV care centers in Nepal with rolling enrollment and treatment from 2015–2018.

### Patients

Patients were recruited from 6 OST sites located in 4 cities: Biratnagar, Dharan, Pokhara, and Kathmandu (3 geographical sites), Nepal (Fig 1). All sites were within areas affected by the severe Gorkha Earthquake that occurred in 2015; Kathmandu sites were most severely hit.

### Personnel

In each city (n = 4) we engaged 2 doctors (2 hepatologists overall and 6 general practitioners), 2 nurses, 1 community focal person, 1 community worker and 1 phlebotomist.

Inclusion criteria were as follows: (i) active HCV infection by rapid and viral load testing; with or without HIV-infection, documented by HIV rapid or viral load testing, (ii) Residing in either Kathmandu, Pokhara, Dharan or Biratnagar and (iii) willing and able to make weekly visits for care and research follow-up, as required for at least 1 year. In phase 2, criteria restricted HIV-infected individuals with (a) undetectable HIV viral load, (b) CD4%>14 (CD4 200), (c) stable ARV regimen for minimum 8 weeks. In phase 3, criteria were modified to include HIV-infection regardless of HIV treatment regimen, CD4 level or HIV viral suppression.

Non-inclusion criteria included: (i) patients unable to provide consent for inclusion in the study; (ii) pediatric patients (age < 18); (iii) any social issues that precluded patients to maintain routine follow-up appointments.

After written informed consent was obtained from all patients, demographic data and biological parameters (CD4 cell count, complete blood cell count, creatinine, liver transaminases) were collected. Fibrosis was assessed using the APRI score and in some cases using transient elastography (Fibroscan). We monitored substance use, HCV treatment literacy, and adherence (by timely follow-up and frequency of consultation with medical staff and social workers). Patients were monitored with lab draw and clinic visit, substance use and depression score at time 1, 2 weeks and then monthly until completion. Monthly or bimonthly support group meetings were conducted at OST sites.

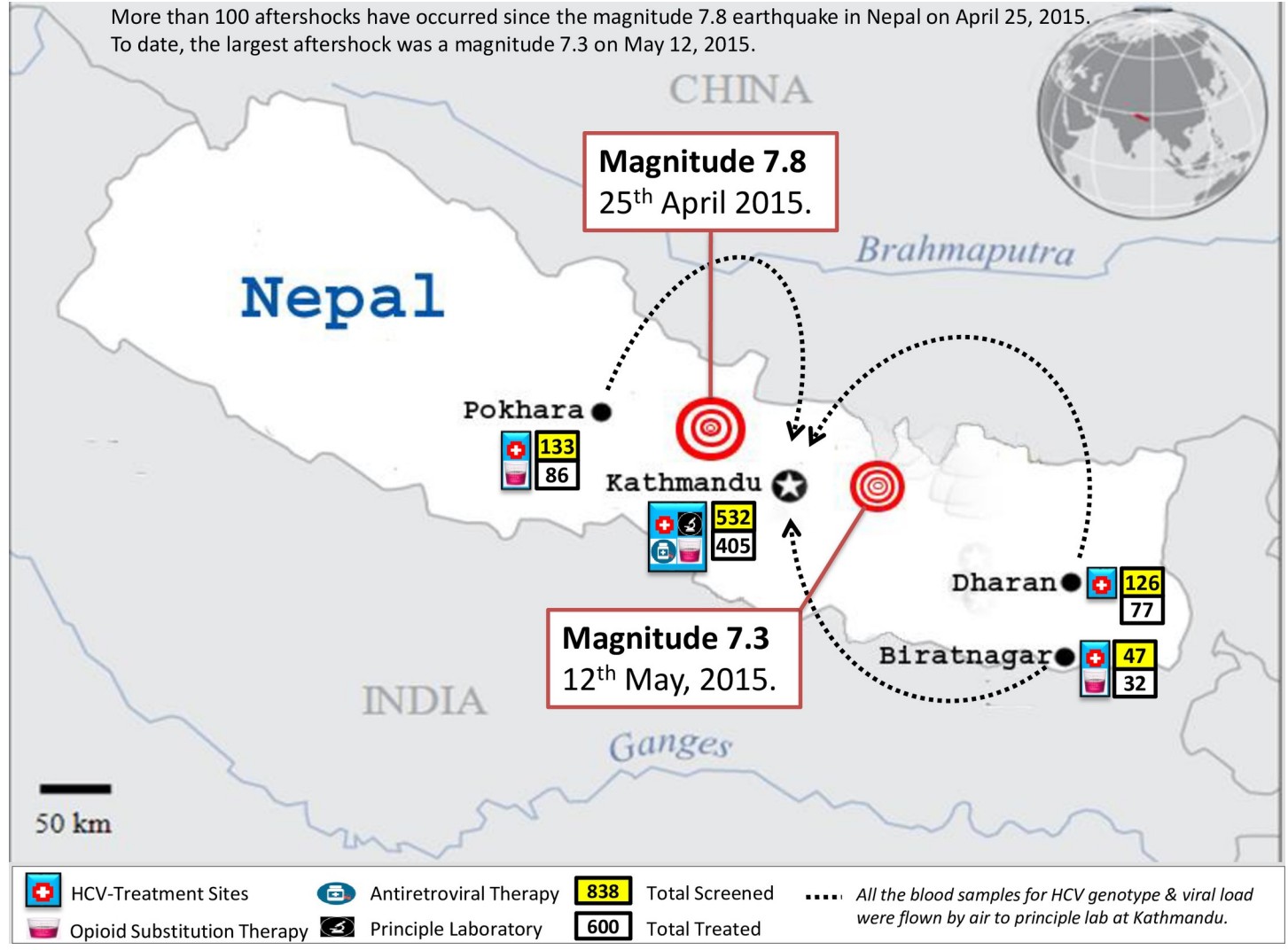

**Fig 1. 2015 Earthquake affected areas and HCV research sites in Nepal.**

### HIV testing

HIV testing was carried out using Point of Care Kit (Determine HIV rapid test, Alere Medical, Japan). If positive results were observed, then second Point of Care Kit (Unigold HIV rapid test, Trinity Biotech, Ireland) was used to confirm. In case of discordant results from two tests, a third and last Point of Care Kit (Tridot HIV Test, J.Mitra & CO, India) test was employed. Clients testing HIV positive were subjected to HIV viral load quantification on SaCycler-96 Real Time PCR System (Sacace Biotechnologies, Italy) using an HCV Real TM-Quant Dx kit (Sacace Biotechnologies, Italy),

### HCV testing

HCV rapid test was by Point of Care Kit (Tridot HCV test, J. Mitra& CO, India). All screened patients, regardless of HCV rapid test results, had HCV quantification performed on a SaCycler-96 Real Time PCR System (Sacace Biotechnologies, Italy) using an HCV Real TM-Quant Dx kit (Sacace Biotechnologies, Italy) able to estimate viral load up to 100,000,000 IU/ml.

HCV Genotyping was performed using the HCV Genotype-Real TM kit (Sacace Biotechnologies, Italy) able to detect HCV genotypes 1a, 1b, 2, 3, 4, 5a and 6. Patients were also tested for hepatitis B virus (HBV) coinfection using hepatitis B surface antigen screening.

Rapid virologic response (RVR) was defined as undetectable HCV RNA at 4 weeks of therapy and sustained virological response (SVR) as undetectable HCV RNA 12 weeks after the end of treatment [8].

### HCV treatment

Three phases according to treatment regimen are distinguished. Phase 1 (interferon-based regimens) is described in S1 Appendix and S1 Table.

Phase 2 treatment began in March 2016, with DAA-based regimens. HCV- and HIV-HCV patients infected with genotype 1 without or with compensated cirrhosis were treated with 12 weeks sofosbuvir/ledipasvir (Natco Pharma, India–approved in Nepal November 2015) and 12 weeks sofosbuvir/ledipasvir/ribavirin, respectively. Patients with genotype 3 received PEG-IFN/ribavirin/sofosbuvir therapy for 12 weeks if they had all optimal predictors (HIV-negative, low fibrosis score, low baseline HCV viral load) and RVR; or sofosbuvir/daclatasvir (Natco Pharma–approved in Nepal March 2016) (12 weeks) or sofosbuvir/daclatasvir +/- ribavirin (24 weeks) (compensated cirrhotics).

Phase 3 treatment started in August 2017. Due to transition to entirely DAA-based treatment (with ribavirin added for cirrhotic patients), the protocol was modified to expand treatment for all patients, regardless of HCV disease status, HIV viremia or drug use. Patients were excluded if they were unlikely to tolerate treatment for medical reasons or inaccessible for follow-up. This phase included those with decompensated cirrhosis. HCV- and HIV-HCV patients infected with genotype 1 with or without compensated cirrhosis were treated with 12 weeks sofosbuvir/ledipasvir/ribavirin and 12 weeks sofosbuvir/ledipasvir, respectively. Patients infected with genotype 3 or another or unknown genotype received sofosbuvir/daclatasvir (12 weeks) or sofosbuvir/daclatasvir +/- ribavirin (24 weeks) (compensated or decompensated cirrhosis).

### Statistical analysis

Qualitative data are presented as numbers and percentages and quantitative variables as median and interquartile range [IQR]. A uni- and multivariable logistic regression analysis was conducted to identify factors potentially associated with SVR12. Variables with a p-value $<0.10$ in univariable analysis were entered into the multivariable model. A p-value$<0.05$ was considered as statistically significant. All analyses were performed using R (R Foundation for Statistical Computing, Vienna, Austria).

## Results

We screened 838 patients who reported hepatitis C infection; most had prior diagnosis based on only HCV Ab testing (Fig 2). Among them, 780 were HCV Ab positive or HCV Ab negative/HCV RNA positive (n = 3); overall 701 (90%) had detectable viremia.

### Phase 2 (Treated March-November 2016)

During phase 2, 116 patients qualified for DAA therapy and 19 patients qualified for IFN-based therapy with sofosbuvir; 135 in total. Median age was 37 years [IQR 33–41] and 97.8% were males (Table 1). The most frequent genotype was 1a observed in 69 patients (51.1%) followed by genotype 3 observed in 56 patients (41.5%). Ten patients had either another or an

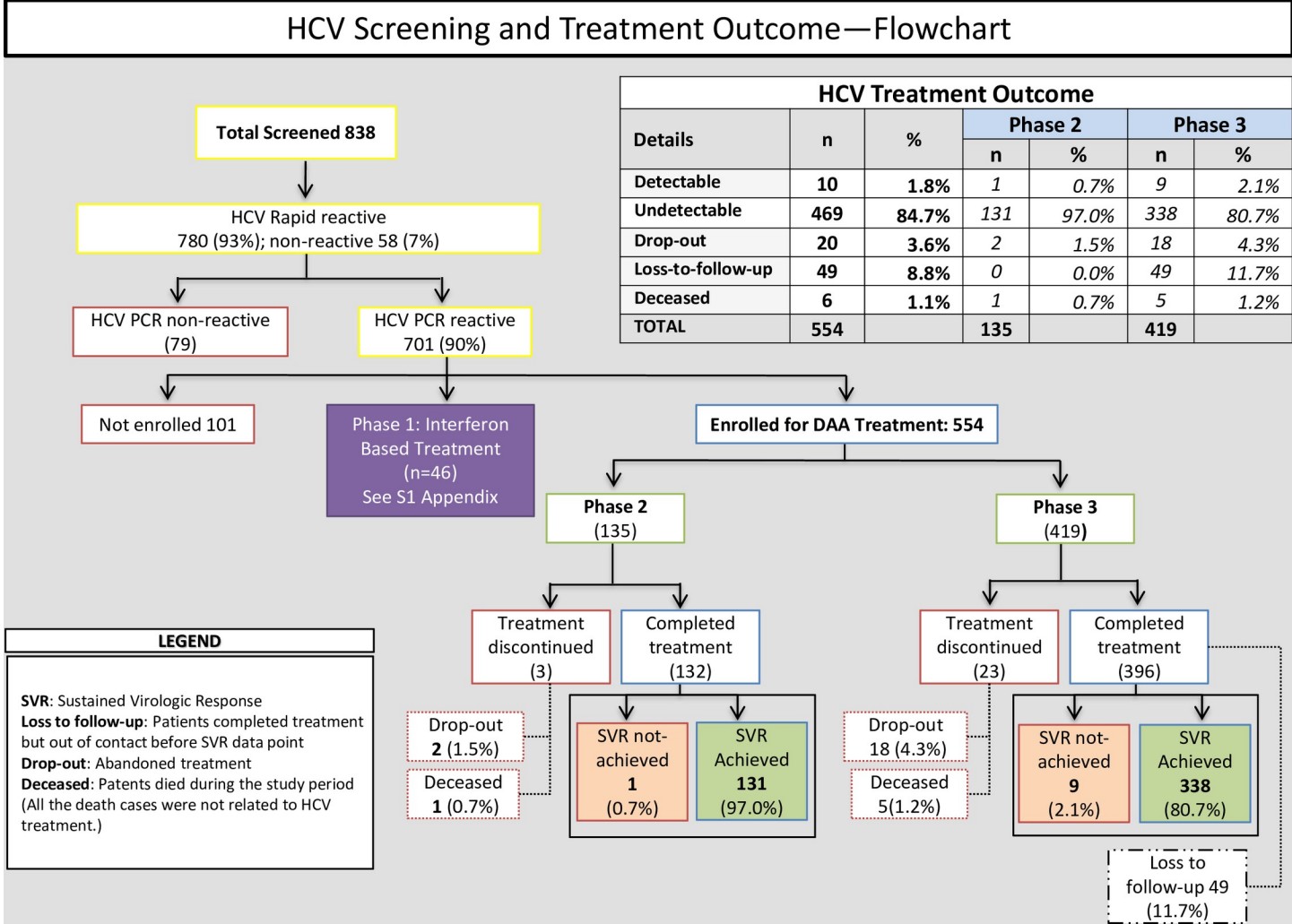

**Fig 2. HCV Screening and Treatment.**

unspecified genotype. Median pretreatment viral load was 5.9 log IU/mL. Sixty-eight patients (50.4%) were HIV-HCV coinfected; three had a detectable HIV viral load. Forty-eight patients (35.6%) had compensated cirrhosis. Patients were treated with sofosbuvir/ledipasvir (44.4%), sofosbuvir/daclatasvir (25.2%), or PEG-IFN/ribavirin/sofosbuvir (14.1%) (Table 2).

None of 135 patients treated exhibited severe adverse events (SAE) related to the treatment protocol. One of 135 had SAE unrelated to treatment protocol, resulting in death. This 41 year-old HIV-HCV coinfected patient with a high CD4 count who received sofosbuvir/ledipasvir died from (TB-negative) pneumonia several weeks after treatment completion. Ten other patients had minor complaints (skin rash, anemia); none required medication change or discontinuation. Three patients dropped out of the protocol before completing treatment; two for drug use relapse. Three patients developed clinical tuberculosis (TB) during (n = 2) or after (n = 1) DAA-based therapy (without interferon). SVR was achieved in 131 patients (97.0%).

**Table 1. Patients' characteristics.** Qualitative data are presented as numbers and percentages and quantitative variables as median and interquartile range [IQR].

| | Phase 2 (n = 135) | Phase 3 (n = 419) |
|---|---|---|
| Site | | |
| Biratnagar | 7 (5.2) | 25 (6.0) |
| Dharan | 12 (8.9) | 55 (13.1) |
| Kathmandu | 71 (52.6) | 309 (73.7) |
| Pokhara | 45 (33.3) | 30 (7.2) |
| Male gender | 132 (97.8) | 399 (95.2) |
| Age, years | 37 [33–41] | 37 [33–42] |
| BMI | 21.3 [19.9–23.5] | 23.1 [21.7–24.5] |
| HIV-HCV coinfection | 68 (50.4) | 175 (41.8) |
| HBV coinfection | 1 (0.7) | 1 (0) |
| APRI score | 1.03 [0.66–1.63] | 0.90 [0.52–1.66] |
| Cirrhosis | 48 (35.6) | 127 (30.3) |
| HCV genotype | | |
| 1a | 69 (51.1) | 163 (38.9) |
| 3 | 56 (41.5) | 230 (54.9) |
| Other | 10 (7.4) | 26 (6.2) |
| HCV viral load before treatment (log IU/mL) | 5.9 [5.3–6.3] | 6.2 [5.5–6.9] |

## Phase 3 (Treated August 2017-January 2019)

A total of 419 patients were treated during phase 3, all treated after the Gorkha earthquake. Median age was 37 years [IQR 33–42] and 95.2% were males. Genotype 3 was observed in 230 patients (54.9%) followed by genotype 1a in 163 patients (38.9%). Median pretreatment viral load was 6.2 log IU/mL. HIV-HCV coinfection was present in 175 patients (41.8%). Among them, 92% had undetectable HIV RNA. Cirrhosis was present in 127 patients (30.3%); 6 with decompensated cirrhosis. Treatment regimens were sofosbuvir/daclatasvir +/- ribavirin combination (61.6%) or sofosbuvir/ledipasvir +/- ribavirin

**Table 2. HCV treatment regimens and treatment response.**

| | Phase 2 (n = 135) | Phase 3 (n = 419) |
|---|---|---|
| **HCV treatment regimen** | | |
| PEG-Interferon+ribavirin | - | - |
| PEG-Interferon+sofosbuvir+ribavirin | 19 (14.1) | - |
| Sofosbuvir+daclatasvir-30mg | 9 (6.7) | 18 (4.3) |
| Sofosbuvir+daclatasvir-30mg+ribavirin | 3 (2.2) | 17 (4.1) |
| Sofosbuvir+daclatasvir-60mg | 33 (24.4) | 113 (27.0) |
| Sofosbuvir+daclatasvir-60mg+ribavirin | 3 (2.2) | 38 (9.1) |
| Sofosbuvir+daclatasvir-90mg | 1 (0.7) | 45 (10.7) |
| Sofosbuvir+daclatasvir-90mg+ribavirin | - | 27 (6.4) |
| Sofosbuvir+ledipasvir | 60 (44.4) | 153 (36.5) |
| Sofosbuvir+ledipasvir+ribavirin | 7 (5.2) | 8 (1.9) |
| **HCV treatment duration** | | |
| 12 weeks | 119 (88.1) | 330 (78.8) |
| 24 weeks | 16 (11.9) | 89 (21.2) |
| **Sustained virological response (SVR)** | | |
| Deceased | 1 (0.7) | 5 (1.2) |
| Detectable | 1 (0.7) | 9 (2.1) |
| Drop-out | 2 (1.5) | 18 (4.3) |
| Out of contact | - | 49 (11.7) |
| Undetectable | 131 (97.0) | 338 (80.7) |

(38.4%). Five patients died, 3 due to pneumonia, and 2 due to liver failure. All deaths were among HIV-HCV coinfected patients.

SVR was achieved by 338 patients (80.7%), whereas nine patients (2.1%) had detectable HCV RNA 12 weeks after the end of treatment. In the remaining patients, no data on SVR was available (Table 2). In the subgroup of phase 3 patients with data on SVR (per protocol analysis), the SVR rate reached 97.4% (338/347). Combining all treatment phases, the per-protocol SVR rate was 97.1% (507/522) and the intent-to-treat (ITT) SVR rate was 84.5% (507/600).

The ITT SVR rate during phase 3 (80.7%) was significantly lower than that during phase 2 (97%; p<0.001).

During phase 3, 49 patients (8.2%) were lost to follow-up prior to SVR time point. Among them, 42% were mobile populations outside the capital; 71% were HCV mono-infected, 45% had cirrhotic liver disease and 18% reported continued drug-use. These patients were targeted by Peer Outreach Workers by word of mouth and phone calls and not located. These patients were disproportionately from the Dharan site representing 34% of the total loss-to-follow-up and 22% of patients at that site vs. loss to follow-up of 3–5% of patients at other sites (p = 0.036 by Chi-2 test). In phase 3, adherence to the protocol was higher for HIV co-infected individuals (92%) than HCV mono-infected patients (85%), (p = 0.011 by Chi-2 test). Funding for OST and Peer Outreach personnel ran out during phase 3, correlating temporally with patient attrition before SVR time point in this phase.

## Phase 2 and 3 analysis

We performed univariable and multivariable analysis to explore factors associated with SVR at 12 weeks (Table 3). We found that treatment in the Kathmandu sites was associated with SVR compared to treatment at the two sites outside of the capitol (OR 3.12 [1.20-8.13]; p=0.020). Substance use was associated with failure to achieve SVR (0.21 [0.10-0.44]; p<0.001). There were more patients reporting substance use during protocol during phase 3 (11%) than in phase 2 (4%). Patients treated in phase 3 were less likely to achieve SVR than those treated during phase 2, (0.08 [0.02-0.23]; p<0.001) a difference that remains after controlling for

**Table 3. Uni- and multivariable logistic regression analysis to predict SVR at 12 weeks.**

| | Univariable analysis | | Multivariable analysis | |
|---|---|---|---|---|
| | OR [95% CI] | p | OR [95% CI] | P |
| Phase (3 vs 2) | 0.13 [0.05–0.35] | <0.001 | 0.08 [0.02–0.23] | <0.001 |
| Site (vs Biratnagar) | | | | |
| Dharan | 0.71 [0.26–1.91] | 0.494 | 0.84 [0.30–2.40] | 0.750 |
| Kathmandu | 2.14 [0.87–5.23] | 0.096 | 3.12 [1.20–8.13] | 0.020 |
| Pokhara | 1.12 [0.41–3.08] | 0.826 | 0.58 [0.19–1.76] | 0.335 |
| Gender (M vs F) | 1.57 [0.57–4.34] | 0.388 | - | - |
| Age | 0.99 [0.95–1.03] | 0.548 | - | - |
| BMI | 1.04 [0.95–1.14] | 0.353 | - | - |
| Cirrhosis (Yes vs No) | 0.64 [0.40–1.04] | 0.071 | 0.70 [0.40–1.22] | 0.205 |
| Genotype (vs 1a) | | | | |
| 3 | 0.76 [0.47–1.25] | 0.282 | - | - |
| Other | 0.77 [0.30–2.00] | 0.594 | - | - |
| HCV viral load (log) | 0.92 [0.74–1.14] | 0.439 | - | - |
| APRI score | 0.96 [0.84–1.10] | 0.562 | - | - |
| HIV coinfection (Yes vs No) | 1.53 [0.94–2.47] | 0.085 | 1.57 [0.92–2.67] | 0.098 |
| Substance use during treatment | 0.36 [0.19–0.68] | 0.001 | 0.21 [0.10–0.44] | <0.001 |

substance use. Demographics, liver disease measured by APRI, cirrhosis and HIV infection status and genotype were not associated with SVR.

## Discussion

This study is the first reporting a community-based HCV treatment experience and the first description of HCV treatment among HIV co-infected individuals in Nepal. We show that overall excellent SVR rates may be obtained in a difficult-to-treat population in a low-income country regardless of the patient's profile, disease severity–including decompensated cirrhosis, and HIV status. The success of the study was made possible by pivoting care off of existing OST sites where clients had access to peer-based support and HIV disease management tools. We demonstrate a way forward to expand access to treatment in Nepal.

In 2015, WHO estimated that only 20% of HCV-infected individuals worldwide were diagnosed and only 7% had started treatment with very large country-dependent variations [9]. A recent systematic review reporting seroprevalence of HBV, HCV, and HIV in Nepal showed an increasing number of studies in recent years illustrating the progressive scientific development by Nepalese researchers and institutions [10]. Remarkably, virologic diagnosis of HCV infection linked to clinical care became available for the first time in Nepal through this protocol. When this study was designed in 2014, persons living with HCV infection in Nepal were desperate. The only source of treatment was by virologic diagnosis in India and self-pay for local IFN-treatment, which was expensive. Most patients "living with HCV in Nepal" had never had a virologic diagnosis or any assessment of fibrosis status. They faced social discrimination and discrimination from work. Rather than waiting for DAAs to emerge in a place where even ARV regimens continue to lag, we proceeded carefully with IFN-based regimens. Restrictions on who could safely be treated with IFN-based therapy limited our treatment cohort (described in S1 Appendix) just as DAAs came on the horizon out of India through collaborative agreements with Gilead. The stakeholders and investigators had a key opportunity to advocate for DAA access in Nepal and were uniquely situated to immediately transition to DAA-based treatment. We expanded the target treatment group to all comers, regardless of fibrosis, HIV disease control and active drug and alcohol use. High cure rates were achievable with DAAs and during a natural disaster even in presence of pejorative prognostic factors such as advanced liver disease (including decompensated cirrhosis) and HIV-HCV co infection. Other studies had demonstrated successful treatment of HCV with DAA in resource-limited settings suggesting that HCV elimination is feasible in such countries [11,12]. The authors stressed that reductions in the cost of antivirals and linkage to social and behavioral health services including substance use disorder treatment were critical to increase retention and adherence to treatment [12].

Our community-based protocol linked to OST sites was remarkably effective at minimizing loss to follow-up with 87% of all patients completing the entire protocol (treatment and follow-up testing).We explored effective aspects of our protocol to describe a Care Pathway moving forward. The majority of patients were more comfortable receiving HCV care at the OST site than the private clinic. Nurses and a community focal person at OST sites were able to guide care and triage to clinicians effectively after an initial clinician visit. Patient feedback optimized the protocol in later treatment stages. We found that monthly visits and bi-monthly support and meeting group (adapted from existing OST and HIV peer meetings), and Peer Outreach through phone calls and home visits optimized participation and minimized patient attrition.

We hypothesize that Kathmandu treatment was protective for several reasons. Access to the lab was optimal in Kathmandu where viral loads were conducted as patients had regular

access. At outside sites, coordinated days for lab draw and plane transport contributed to missed testing. This problem may be addressed by Hep C core Ag for SVR for sites without access to virologic testing. There was a higher rate of geographic mobility for work outside of the capitol and thus more difficulty maintaining contact. Patients perceived that confidentiality was more threatened in smaller communities where more care was also conducted in the clinical setting and stated that this challenged follow-up. There was a more comprehensive network of organizations working for IVDU in Kathmandu which helped in tracking patients. Improved linkages between HCV care and existing organizations is likely a key to success for program expansion.

During phase 3 the government of Nepal introduced a campaign mid protocol to actively incarcerate people with pending legal cases from the distant past who were arrested and remanded because of their previous cases related to drug use and other petty crimes. Only 1 incarcerated patient was able to remain in the study after extensive work on behalf of the investigators which highlights HCV treatment in jail as a need area in Nepal, where the rate of drug users in prison populations is >10% and as high as 28% [13]. Substance use was higher during phase 3 and independently associated with dropping out of the protocol.

Subsequent to initiation of this study, data has shown that genotypes 3 non-3a are associated with natural prevalence of NS5A polymorphisms. To date, genotype 3 subtype analysis has not been done in Nepal and was not feasible in our system which we recognize as a weakness. Eight of 10 patients with virologic failure and those lost to follow-up before SVR will benefit from genotypic subtype and resistance testing, after Hep C core Ag testing to identify replicating virus for the latter. Advocacy to obtain glicaprevir/pibrentasvir–not currently available in the region, may be important if genotype non-3a is described and as DAA use is more prevalent.

Initially, we observed that waiting to optimize HIV treatment regimen and viral suppression substantially delayed HCV treatment and contributed to patient attrition. Ultimately we promoted HCV treatment regardless of HIV regimen and control status with excellent outcomes, as shown previously [14]. In going forward in Nepal, we advocate for HCV treatment regardless of HIV control based on these experiences.

There were overall minimal poor outcomes considering this setting and coinfections. There were 6 deaths; all occurred in DAA-treatment groups and among HIV coinfected patients and predominantly among those with cirrhosis (5 of 6); 4 of 6 with concurrent drug use; 4 of 6 due to pneumonia. This highlights a group that may require closer attention and more careful follow-up. There were 3 cases of TB pneumonia during DAA therapy though there is no known association. This is an area for research.

Overall completion and retention in care is more remarkable considering unique challenges in Nepal during this protocol. Nepal faced the severe Gorkha earthquake on April 25, 2015 with subsequent quakes and aftermath during this protocol. We estimate that this earthquake directly affected at least 20% of our enrolled population and 100% of patients were socially affected. All patients were enrolled at that time with none from phase 2 and 3 having initiated treatment. All collected samples were retained in appropriate conditions with emergency power supplies despite the substantial breakdown in the electrical grid in the Kathmandu and elsewhere in the weeks immediately following the earthquake. Some patients who had not yet initiated treatment or not yet obtained SVR data were lost to follow-up due to migration after the event. Due to delays, largely related to logistics surrounding the earthquake, the timeframe of our protocol was shifted. Phase 2 treatment was so delayed that phase 3 encompassed a 3-times larger cohort (413 vs. 135). In addition this resulted in final SVR data collection extending beyond the funding mechanism. The volume and delays likely contributed to the high attrition before data collection during phase 3.

One of the main barriers to DAA-based treatment access in resource-limited countries is their elevated cost. Moreover, even though DAAs have become available at highly subsidized cost (approximately 800–1000 USD, near the average annual income in Nepal), DAAs remain cost prohibitive for most Nepalese [7]. In this context, the arrival of low-cost generic drugs will be most effective in the presence of HCV treatment as a national agenda and with government or nongovernmental funding [15]. Government supported treatment has been discussed for a finite HIV-HCV co-infected cohort in Nepal but there are no concrete plans for the HCV mono-infected population, or for risk groups such as diabetics, prisoners and cirrhotics at this time.

Based on our experience, a Care Pathway for HCV care expansion in settings like ours may consider: OST-based care with a designated community focal (non-medical) leader and intermediate level provider with doctor oversight, broad serologic testing for HCV with reflex Hep C core Ag and APRI, Hepatitis B Ag and HIV testing, monthly visits (clinical with substance use and depression screening). Funded bimonthly support group and peer counseling by paid staff throughout care and until SVR with more intensive peer support for active substance users should be prioritized. Hep C core Ag testing at 12 weeks after completion for SVR rather than virologic testing would optimize outcomes analysis at rural sites. The reliability and durability of earlier SVR is an important area for research. Clinical care will be simplified with pan-genotypic drug.

The 2016 WHO objective of "eradicating" HCV by the year 2030 was later determined to be a too ambitious target and objectives were revised. Some of the new objectives include the detection of 90% of patients chronically infected with HCV and treatment of 80% of patients chronically infected [3,16]. These screening and treatment rates still seem unrealistic, especially for resource-limited countries [17]. A substantial treatment scale up is needed especially in IVDU to control the HCV epidemic [18]. Scale up based on decentralization of HCV treatment with DAAs prescribed by non-specialized personnel at primary care settings was shown to be effective in terms of SVR12 [19]. Our network linking OST clinics in distant parts of the country to higher level testing and care in the capital is a model for further initiatives to treat individuals living with HCV in Nepal and may be applied elsewhere to help achieve these goals.

## Supporting information

**S1 Appendix. Phase 1, Interferon-based treatment.**
(DOCX)

**S1 Table. Characteristics of patients treated by PEG-Interferon+ribavirin during phase 1 (n = 46).**
(PDF)

## Acknowledgments

We dedicate this work to our colleague, Nabin Tulachan. We are grateful to Natco Pharma and Dr. Betty Chiang from Gilead for donation of DAAs.

## Author Contributions

**Conceptualization:** Sudhamshu KC, Holly Murphy, Sameer Dixit.

**Data curation:** Apurva Rai, Anjay KC.

**Formal analysis:** Holly Murphy, Apurva Rai, Pierre Pradat.

**Funding acquisition:** Ujjwal Karmacharya, Philippe Creac'H.

**Investigation:** Sudhamshu KC, Holly Murphy, Sameer Dixit, Apurva Rai, Bickram Pradhan, Marie Lagrange-Xelot, Amélie Dureault, Santosh Panthi, Nabin Tulachan, Prawchan KC, Rajesh Rajbhandari, Andrew B. Trotter.

**Methodology:** Sudhamshu KC, Holly Murphy, Sameer Dixit, Apurva Rai, Ujjwal Karmacharya.

**Project administration:** Ujjwal Karmacharya, Philippe Creac'H.

**Resources:** Apurva Rai, Niyanta Karki, Santosh Panthi, Nabin Tulachan, Anjay KC.

**Supervision:** Sudhamshu KC, Holly Murphy, Sameer Dixit, Jörg Gölz, Christian Trépo.

**Visualization:** Apurva Rai, Pierre Pradat.

**Writing – original draft:** Holly Murphy, Pierre Pradat.

**Writing – review & editing:** Sudhamshu KC, Holly Murphy, Sameer Dixit, Apurva Rai, Bickram Pradhan, Marie Lagrange-Xelot, Niyanta Karki, Amélie Dureault, Ujjwal Karmacharya, Prawchan KC, Anjay KC, Rajesh Rajbhandari, Andrew B. Trotter, Jörg Gölz, Pierre Pradat, Christian Trépo, Philippe Creac'H.

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
