## [Decision Letter · Decision Letter 0]

6 Aug 2020

Dear Dr Pradat,

Thank you very much for submitting your manuscript "Hepatitis C (HCV) therapy for HCV mono-infected and HIV-HCV co-infected individuals living in Nepal" for consideration at PLOS Neglected Tropical Diseases. As with all papers reviewed by the journal, your manuscript was reviewed by members of the editorial board and by several independent reviewers. In light of the reviews (below this email), we would like to invite the resubmission of a significantly-revised version that takes into account the reviewers' comments. 

This article is a welcome contribution, highlighting a programme for HCV treatment among a difficult to reach and low income population in Nepal. Please note the comments from reviewers below. 

We would request that you remove the interferon-based analysis (phase 1) from the main article and abstract as this includes a small number of patients and we consider these data obsolete and unlikely to be applicable to future programmes. This is particularly important when considering generalisable lessons from the programme to other settings and in the consideration of factors associated with failure to achieve SVR. These data from phase 1 could instead be included in a supplementary appendix. 

We also ask that you include a focus on lessons for improving the care pathway since the (successful) management of HCV using DAAs is already established in several low and middle income settings.

Finally figure 2 should be revised to improve readability.

We cannot make any decision about publication until we have seen the revised manuscript and your response to the reviewers' comments. Your revised manuscript is also likely to be sent to reviewers for further evaluation.

Sincerely,

Alexander Stockdale, PhD MRCP

Deputy Editor

Reviewer's Responses to Questions

**Results**

-Does the analysis presented match the analysis plan?

-Are the results clearly and completely presented?

-Are the figures (Tables, Images) of sufficient quality for clarity?

Reviewer #1: -Does the analysis presented match the analysis plan? yes

-Are the results clearly and completely presented? Some additional information is requested

-Are the figures (Tables, Images) of sufficient quality for clarity? No for Figure 1.

Reviewer #2: In my point of view, the authors should show what is the best way to get the highest SVR12.

Frequency of visits ? why did you choose a visit every 2 weeks and not every week.

How do you explain the SVR12 difference between phase II and phase III ?

Can you do a multivariate analysis to predict the SVR12. The number of patients included is high and the number of variables to study is roughly ten to twelve : genotype, viral load, cirrhosis, alcohol, drug injection during the treatment, centre, phase, distance from the center, housing conditions, level of study, treatment during the earthquaker period......

**Summary and General Comments**

Reviewer #1: The authors report the history of HCV treatment in Nepal from 2014 to 2018 of a total of 600 treated patients, out of 830 individuals screened for HCV, with three types of treatment, according to the time period: IFN/RBV, DAAs/RBV and DAAs alone. The results show a high rate of response (even in the first phase) and warrant further program for a mass treatment of HCV in Nepal. Some considerations should be addressed before considering publishing this manuscript.

Major comments

1. The use of rapid tests for identifying HCV or HIV carriers might be a concern in terms of sensitivity. The authors should discuss this issue.

2. A high rate of SVR was obtained even with the use of IFN/RBV, although the most abundant genotype is 1. On the other hand, it has been described that some G3b may harbor DAAs natural mutations, and this subtype circulates in the region. Is there any additional information of the genotype/subtype of HCV in the non-responders? The authors should also discuss the genotype/subtype distribution in the country. Even if the present HCV treatment is pangenotypic, some focal diversities might play a role in reducing the high rate of SVR.

3. No statistical comparison is performed between the SVR rates of the different phases, if the population groups are comparable.

Minor comments

4. The reference of the earthquake of 2015 in Figure 1 is somewhat confusing. It would be preferable to include a comment on the earthquake in the text in page 6 lines 105-107, and include in the figures only the geographic location of the centers and the center.

5. Page 6: line 111: the inclusion criteria should include HCV positivity by rapid test and a reference to Figure 2.

6. Page 7 line 143: instead of HCV testing, it is HCV RNA detection and genotyping.

7. Page 6: the discussion of the cost of DAAs should be revised in light of availability of generic drugs.

Reviewer #2: I think you should focus on care pathway.

How can you help other countries that want to reproduce your experience ?

Reviewer #2: -This paper analyses the feasibility of the treatment of IVDU with hepatitis C in Nepal.

-It has been previously shown that the results of hepatitis C treatment in low income countries is as good as in high income countries (Bull World Health Organ. 2012 Jul 1;90(7):540-50. doi: 10.2471/BLT.11.097147.)

-I then think that the paper should focus more on the care pathway than on SVR12

-We can easily imagine that the results will be as good with the DAAs.

-Then the per-protocol results are not very useful.

-We want to know what are the risk factors that reduce intention to treat SVR12

-How many people have worked on the project ? How many people have been hired (nurses....) ?

-I am not sure that the group 1 is very useful as nobody treat hepatitis C with this treatment nowadays.

-Can you tell us when the different phases started and finished ? When the earthquaker and its consequencces started and finished.

-The number of references in the bibliography section is very low. Is there another experience in another countries ? in IVDU ? what have we learned from the egyptian experience ?

-What can we expect from the pangenotypic treatment (to add in discussion section)?

**Conclusions**

-Are the conclusions supported by the data presented?

-Are the limitations of analysis clearly described?

-Do the authors discuss how these data can be helpful to advance our understanding of the topic under study?

-Is public health relevance addressed?

Reviewer #2: -80% of SVR12 in phase III in ITT analysis cannot be considered a good result.

-There is no data explaining the difference between the phases II and 3 ? Is it due to the earthquaker ?
---

## [Editor Report · Decision Letter 1]

30 Oct 2020

Dear Dr Pradat,

We are pleased to inform you that your manuscript 'Hepatitis C (HCV) therapy for HCV mono-infected and HIV-HCV co-infected individuals living in Nepal' has been provisionally accepted for publication in PLOS Neglected Tropical Diseases.

We request that you remove the reference to the supplementary appendix within the abstract, to modify the sentence: "Patients treated with interferon-based regimens (phase 1) are discussed in S1 Appendix (n=46)". Instead of requesting a further minor revision and resubmission, I would ask that this minor change is made at the proof stage.

Best regards,

Alexander Stockdale, PhD MRCP

Deputy Editor

---

## [Editor Report · Acceptance letter]

28 Nov 2020

Dear Dr Pradat,

We are delighted to inform you that your manuscript, "Hepatitis C (HCV) therapy for HCV mono-infected and HIV-HCV co-infected individuals living in Nepal," has been formally accepted for publication in PLOS Neglected Tropical Diseases.

Best regards,

Shaden Kamhawi

co-Editor-in-Chief

Paul Brindley

co-Editor-in-Chief
